# Application of Ultra-High-Performance Concrete in Bridge Engineering: Current Status, Limitations, Challenges, and Future Prospects

**S. Abdal [1,2], Walid Mansour [3], Ibrahim Agwa [4], Mohammed Nasr [5], Aref Abadel [6], Yasin Onuralp Özkılıç [7] and Mahmoud H. Akeed [8,*]**

1 Department of Civil Engineering, College of Engineering, University of Duhok, Duhok 42001, Iraq
2 Department of Civil Engineering, College of Engineering, Nawroz University, Duhok 42001, Iraq
3 Department of Civil Engineering, Faculty of Engineering, Kafrelsheikh University, Kafrelsheikh 33516, Egypt
4 Civil and Architectural Constructions Department, Faculty of Technology and Education, Suze University, Suez 41522, Egypt
5 Technical Institute of Babylon, Al-Furat Al-Awsat Technical University, Babylon 51015, Iraq
6 Department of Civil Engineering, College of Engineering, King Saud University, Riyadh 11451, Saudi Arabia
7 Department of Civil Engineering, Necmettin Erbakan University, Konya 42090, Turkey
8 School of Civil and Environmental Engineering, University of Technology Sydney (UTS), Sydney 2007, Australia
* Correspondence: mahmoud.akeed89@gmail.com

**Abstract:** Ultra-high-performance concrete (UHPC) is a form of cementitious composite that has been the most innovative product in concrete technology over the last three decades. Ultra-high-performance concrete has been broadly employed for the design of numerous forms of construction owing to its excellent mechanical characteristics and durability, and studies on its behavior have grown fast in the last decades. While the utilization of ultra-high-performance concrete in bridge engineering (BE) is limited owing to its high costs, little is recognized about the utilization of UHPC in various BE elements. As a result of these issues, a comprehensive review of the current UHPC development trends should be conducted to determine its present state and perspective. This study presents a review of the state-of-the-art UHPC applications in BE. This review also discusses the current status, limitations, challenges, and areas for the further investigation of UHPC in BE. The aim of this research to help various construction stakeholders understand the distinctive characteristics, benefits, and barriers to the broad utilization of ultra-high-performance concrete applications. The understanding of UHPC will aid in increasing its entire market share in both the national and worldwide building sectors.

**Keywords:** UHPC; applications; bridge engineering; limitations; challenges; future prospects

## 1. Introduction

Ultra-high-performance concrete (UHPC) is an improved cementitious and fibrous concrete with high compressive strength (CS) (120–250 MPa) [1], tensile strength (15–20 MPa) [2], particle packing density (0.825–0.855) [3], and exceptional durability [4]. UHPC has three hundred times the energy absorption and ductility of high-performance concrete and three to sixteen times the compressive strength of normal concrete [5]. Owing to its excellent toughness and ductility under strain, as well as its remarkable mechanical characteristics, UHPC is often recognized as the material of choice for seismic design reasons [6]. UHPC is a viable alternative for improving infrastructure and the long-term viability of construction facilities [7].

Ultra-high-performance concrete is prepared with a low $w/c$ ratio, often between 0.15 and 0.25 [8]. Owing to the low water volume, high-range water reducer agents are necessary to enhance the packing of the particles in the composite material, leading to

a higher workability and fluidity of the mix [9]. As cementitious binders, PC and silica fume are often employed in UHPC production scenarios [10]. The worldwide ultra-high-performance concrete market is predicted to increase at a multifactor yearly growth rate of nearly seven percent between 2018 and 2022, as shown in Figure 1. As the high greenhouse gas inventory of PC has become a global issue, the need for more sustainable cementing binders has advanced significantly [11–13].

The mechanical characteristics of UHPC make it an ideal material for applications (apps) where strength is a fundamental design objective, and concrete structural component sizes can be reduced to make them thinner, smaller, and more visually appealing [14,15]. It is typically made up of PC, silica fume, fine aggregate, a higher-range water-reducing ingredient, and fibers. UHPC could be an appropriate material for concrete structures exposed to harsh environments [16]. UHPC is commonly employed in ultra-high-rise buildings, BE design, and long-span structures [17]. In harsh climate exposure or outdoor conditions, UHPC reduces service cycles, and the enhanced durability and longevity extends the lifecycle [18]. By utilizing synthetic fibers in UHPC, it is feasible to attain ultra-high CS and high tensile ductility in concrete materials [15,19]. Although UHPC progress and tech have been comprehensively studied and recognized from the micro- to the macro-level, its wide marketing remains difficult due to the high costs and complicated production process [12,20–22]. The problematic production process is mostly because of the usage of too many components, which leads to high prices and difficult handling. UHPC tech provides a different product that allows infrastructure developers to broaden their service offerings and product [4]. This tech's main concept is the introduction of systematic ways to solve the inherent shortcomings in traditional concrete; for example, one of the sustainable ways to produce UHPC can be performed using geopolymer technology [23,24]. This innovative concrete is superior since it is more ductile, with a high capability to deform and support flexural and tensile loads even after initial cracking forms [25–28]. UHPC's improved performance characteristics are the consequence of the improved bonding optimization and mineral matrix microstructural characteristics among the concrete matrix elements [29,30].

Since its inception more than two decades ago, UHPC has attracted increasing attention from the construction industry, with attention on the following: construction BEs [31], damaged concrete components [32], skyscrapers, unique architectural designs [33], offshore constructions, facilities related to the oil and gas industry apps [34], vertical elements (for instance, windmill towers and wind turbines) [7], overlay materials [35], and hydraulic structures [36]. UHPC is widely utilized in all of these industries, as well as road and BE construction [16]. UHPC is especially well suited for BE construction in difficult situations since its composites need less repair throughout their lifecycle and have excellent strength [37]. Ultra-high-performance concrete is also a preferred strategy for BE construction in high-traffic locations since it supports stronger and longer spans, leading to more usable space. However, the quality of the materials utilized and the accuracy with which they are produced have a considerable impact on the functioning of ultra-high-performance concrete [38]. Another factor that leads to carbon dioxide emissions is the high cement volume of ultra-high-performance concrete, which raises ecological concerns [3,39–42]. As a cement substitute, SF with fillers (for example, limestone and quartz powder) can substantially increase the workability of ultra-high-performance, fiber-reinforced concrete and the efficacy of the steel fibers in the material [43]. Moreover, fillers can lower the volume of the microsilica needed, which is essential for ultra-high-performance, fiber-reinforced concrete in terms of energy, costs, and ecological effect [43]. As SF has a broad-range of chemical and physical characteristics, depending on its source, more standardization and study are required [43,44].

Furthermore, UHPC provides a diverse product range that can be employed in big projects and infrastructure, including highways, federal roads, BEs, marine facilities, water conservation facilities, pre-cast buildings, and military facilities [45]. As several service infrastructures and facilities around the world deteriorate, innovative UHPC strategies, such as prefabricated BE components [46], UHPC BE overlays, seismic columns [47], piles [48],

BE girders [49], link slabs [50], cladding [51], and waffle deck panels [52], are achieving acceptance and popularity in the construction industry. Owing to UHPC's advanced progress, architects and designers can now introduce structural and decorative punctured facades in mesh designs or lattice styles, ultrathin [53], lightweight panels [31], and full facades with multifaceted forms, textures, curvatures, and puncture levels exceeding 50 percent [25]. This substantial advancement seems to be limited, though, by a lack of knowledge of the manufacturing processes and raw materials, the restricted design codes, and the high production costs [54].

Nonetheless, the usage of UHPC as a prospective and novel material is earning traction across a wide range of stakeholders, including scientists and construction companies [49]. Nevertheless, numerous hurdles limit the widespread utilization of UHPC. Some examples include significant spalling and shrinkage strains, large volume production techniques with limited workability, and undetermined durability following the progress of long-term concrete cracks [55–57]. Because of a lack of knowledge in the industry, ultra-high-performance concrete experts confront extra hurdles in imparting hands-on expertise to concrete industry professionals so that they can be well versed in the implementation of complex concrete techniques [58].

The aim of this review is to critically analyze the previous investigations on the utilization of UHPC in BE, as well as the present challenges restricting its broad adoption. Knowledge gaps have been found, as have the investigation needs. A study of the state of knowledge is essential since the investigation of UHPC involves a grasp of concrete materials science as well as structural and BE ideas. This review can assist scientists, designers, and practitioners in expanding the usage of ultra-high-performance concretes in advanced apps.

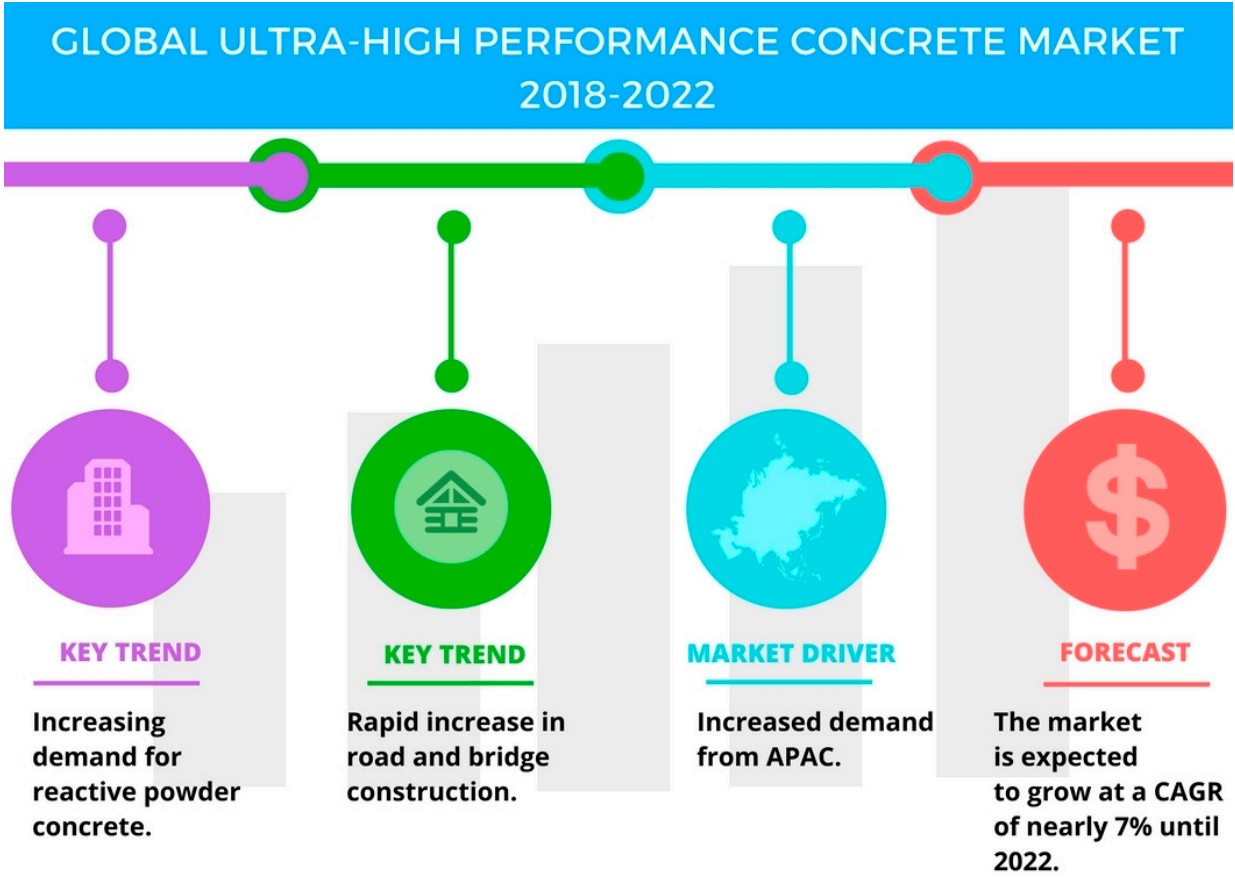

**Figure 1.** Global UHPC Market [59].

*Development of UHPC*

Concrete is the world's most frequently utilized human-made product, and it will carry on being highly popular in the near future. The world's manufacture of concrete is expected to be over six bcm/year, with China currently accounting for almost 40 percent of global concrete manufacturing [60,61]. Concrete's outstanding characteristics, including its durability and strength, low costs, and capacity to be poured into different shapes, have made it the most well-known and important construction material. Concrete is often utilized owing to its strong CS [62,63]. Substantial progress has been achieved in the area of concrete construction during the last several years. In the 1930s, significant scientific efforts to enhance concrete CS started [64–67].

During the 1960s, concrete tech evolved slowly, with maximum CSs varying from 15 MPa to 20 MPa. Concrete's CS grew from 45 MPa to 60 MPa over a ten-year period. Because of the technical obstacle of the present water reducer, concrete strength peaked at roughly 60 MPa in the early 1970s. The present water reducer was unable to further lower the w/b ratio at the time [68,69]. It was found in the 1980s that high-range water reducers, known as superplasticizers, could be utilized to progressively lower the w/b to 0.30. Lowering the w/b below 0.16 was supposed to be unlawful until Bache [70] demonstrated that it was possible to do so with large dosages of silica fume and superplasticizers. A concrete CS of up to 280 MPa was attained by utilizing compacted granular materials by regulating the grain size distribution of the granular skeleton. As a consequence, a material with the fewest flaws, such as interconnected pore spaces and micro-fractures, was advanced to attain maximum durability and strength.

These technical breakthroughs, together with a fundamental understanding of low porous materials, have led to the formation of ultra-high-performance Portland composition materials with enhanced mechanical properties. Usually, the progress of UHPCs can be categorized into four stages: before the 1980s, after the 1980s, after the 1990s, and after the 2000s. Moreover, Figure 2 presents a summary of the historical progress patterns of UHPC.

Due to a lack of the current tech prior to the 1980s, UHPC production was restricted to the lab and needed specialized techniques such as heat curing and vacuum mixing. During this time, scientists experimented with several methods to produce more compact and denser concrete in order to boost its strength. Vacuum mixing coupled with temperature curing has been shown to raise the CS of concrete to 510 MPa [71,72]. Although a high CS of concrete was obtained, the preparation was energy-intensive and time-consuming [73–76].

In the early 1980s, microdefect-free cement was advanced [77]. In the microdefect-free cement process, polymers are employed to seal the pores and eliminate any flaws in the cement paste. Specific production conditions, such as the material being laminated by passing it through rollers, are required for this technique. Microdefect-free cement concrete has a CS of 200 MPa. Nevertheless, its apps have been restricted owing to the complicated preparation procedure, the high costs of the raw ingredients, the brittleness, and the substantial creep [77]. Following the launch of microdefect-free cement, Bache [70] advanced dense silica particle cement (dense silica particle cement). Dense silica particle cement, unlike microdefect-free cement, does not need rigorous manufacturing conditions to be produced. Dense silica particle cement faults were eliminated by increasing the particle packing density. Dense silica particle cement concrete is cured with pressure and heat and has a high concentration of SF and SP. Dense silica particle cement has a maximum CS of 345 MPa. These materials grow increasingly brittle as their ultra-high strength increases. Steel fibers were added to dense silica particle cement concretes in the 1980s to assist in enhancing their brittleness. This form of steel fiber-augmented concrete is a completely different material. It possesses ultra-high strength, a very thick microstructure, excellent ductility, and excellent durability. Slurry-infiltrated fiber concrete (SIFCON) and compact reinforced composites (CRC) are two significant instances of what transpired immediately after dense silica particle cement. Both slurry-infiltrated fiber concrete and compact reinforced composites offer remarkable durability and mechanical characteristics. Nevertheless, both slurry-infiltrated fiber concrete and compact reinforced composite

slurry-infiltrated fiber concrete have workability issues that limit in situ implementations due to a lack of efficient high-range water reducers [78,79].

In the 1990s, Richard and Cheyrezy [80] utilized components with reactivity to produce reactive powder concrete (RPC) and higher fineness via thermal treatment. Reactive powder concrete is a key advancement in the progress of UHPCs. Its idea depended on the very dense arrangement of numerous particles. Reactive powder concrete is the most often utilized kind of UHPC in the field and laboratory experiments owing to its very high cement content, high binder concentration, extraordinarily low water-to-cement ratio, and utilization of fine quartz powder, SF, SP, quartz sand, and steel fibers [42]. To increase the matrix's consistency, the coarse particles are eliminated. The CSs of reactive powder concrete range from 200 MPa to 800 MPa. Table 1 illustrates the typical mechanical parameters and reactive powder concrete composition provided by Richard and Cheyrezy [80]. Unlike its predecessors, reactive powder concrete is very consumer-oriented. This property of workability is an advantage and the most significant criterion for large-scale cementitious material apps. The first reactive powder concrete UHPC, known as Ductal+, was released in the late 1990s. The world's first reactive powder concrete structure was developed in 1997 for a pedestrian BE in Sherbrooke, Canada [81]. It was the first time reactive powder concrete was employed to build the complete framework. Despite the effectiveness of reactive powder concrete structures, the apps are still restricted owing to the production costs and the prohibitively high cost of the materials.

**Table 1.** Typical mechanical parameters and reactive powder concrete composition of UHPC were provided by Richard and Cheyrezy [80].

| | Ingredient in the Manufacture (kg/m$^3$) | | | | | | | | | |
|---|---|---|---|---|---|---|---|---|---|---|
| | PC | Ground Quartz (d50 = 10 mm) | Fine Sand (150–600 mm) | Total Water | Silica Fume | Steel Fibers | Superplasticizer (Polyacrylate) | Heat Treatment | CS (MPa) | Flexural Strength (MPa) |
| reactive powder concrete 200 | 955 | / | 1051 | 162 | 239 | 168 | 15 | 20 C/90 C | 170–230 | 25–60 |
| reactive powder concrete 800 | 1000 | 390 | 500 | 190 | 230 | 630 | 19 | 250 C–400 C | 490–680 | 45–102 |

Since the year 2000, wide development has occurred in the progress of UHPCs. The engineers recognized that as concrete tech grew, advanced concrete should have more useful functions than high strength, which resulted in the names UHPC and ultra-high-performance, fiber-reinforced concrete [82]. A broad range of novel concrete formulations has been advanced to meet an increasing number of apps. Several scientists are now developing sustainable UHPC formulations with the goal of lowering both the initial and the material costs [83]. Supplementary cementitious materials such as SF, fly ash, rice husk ash, ground granulated blast furnace slag, and others [40,79,84], are employed to substitute part of the cement in the progress of sustainable UHPC and to minimize its current cement consumption. It has also been reported that UHPC could be synthesized by utilizing conventional temperature curing without compromising its characteristics. UHPC apps are becoming more common as ecologically friendly UHPC becomes more affordable. Since the early 2000s, numerous nations have been interested in different UHPC apps. UHPC has been employed to build several structures in France, including BEs, slabs, and facades [85]. UHPC is also being employed to repair and maintain transportation infrastructure in the United States [49]. Considerable efforts have been made in Australia to produce UHPC for BE construction [86]. UHPCs have mostly been employed for in situ structural reinforcement in Switzerland [87]. UHPC BE projects have been erected in Spain and the Netherlands [88]. In Malaysia, UHPC has been employed for BE construction as part of a sustainable BE building plan. Since 2010, a total of 113 UHPC BEs have been completed or advanced in Malaysia [89].

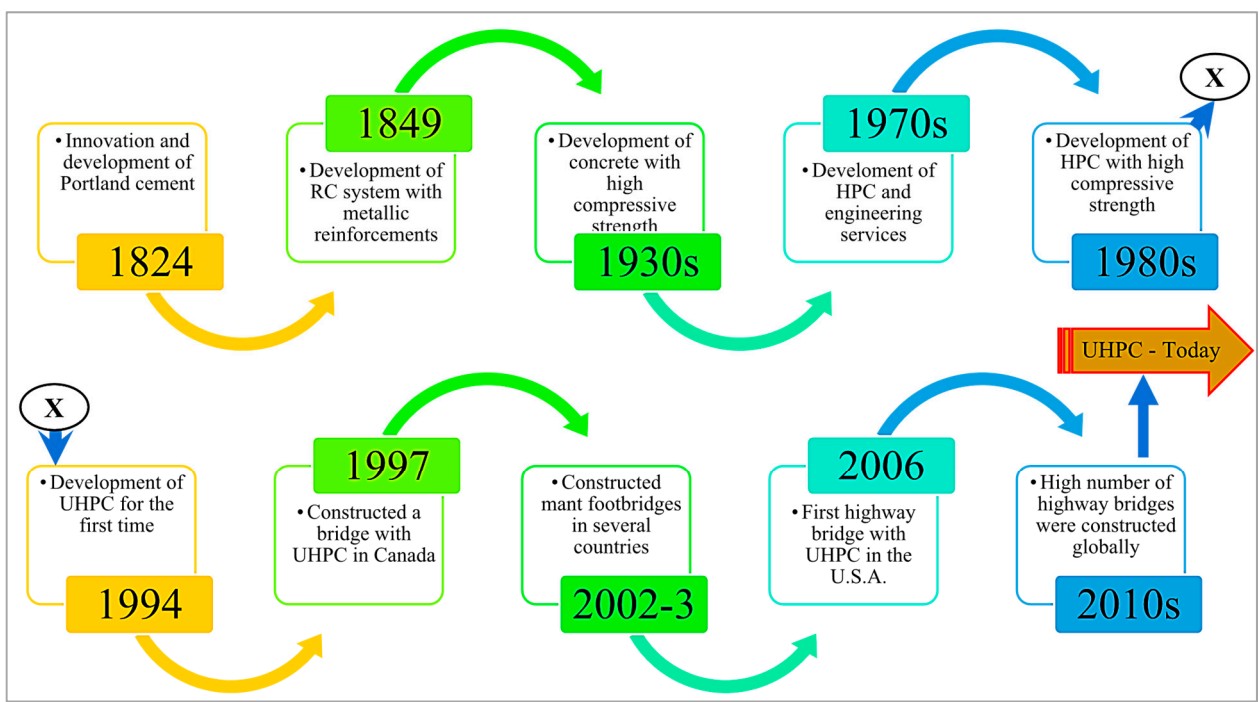

**Figure 2.** UHPC's historical progress patterns [90].

## 2. Applications of UHPC in Bridge Engineering

In recent years, the utilization of UHPCs in BE has steadily improved, with high achievements so far. In Malaysia alone, twenty-six UHPC BEs were constructed in 2014 and 2015 [63]. As another example, UHPCs became commercially available in the U.S. in 2000 and were utilized to build the first UHPC girder BE, known as the Mars Hill BE, in Wapello County, Iowa. Furthermore, according to a recent Federal Highway Administration (FHWA) study, about 200 BE construction projects from 2006 to 2018 utilized a UHPC proprietary mix at a given scale, whether maintaining an existing BE or constructing a new BE within the state DOT BE network. Table 2 presents completed UHPC projects worldwide. In addition, Figure 3 depicts the number of UHPC BEs built in North America between 2006 and 2016.

**Table 2.** Completed UHPC projects worldwide [53].

| Year | App | Location | Advantages |
|------|-----|----------|------------|
| 1997 | Pedestrian BE | Sherbrooke, Canada | The first UHPC structure. |
| 2004 | Foot BE | Seonyu, Seoul, South Korea | Reduced-segment arch BE. |
| 2004 | Roof | Shawnessy LRT Station, Canada | Simple to build and requires very little maintenance work, lightweight. |
| 2005 | Road BE | Bourg-lesValence, France | Steel reinforcing costs are reduced by 90 percent. Lighter construction with a 66 percent weight reduction over CC. |
| 2006 | Road BE | Mars Hill BE, United States | The first UHPC highway BE in the United States had a simple structure. There is no shear reinforcement. |
| 2013 | Column and Façade | MUCEM, Marseille, France | Y-formed column with a 'Transparent' façade. |
| 2013 | Roof and Façade | Jean Bouin Stadium, Paris | Pre-cast UHPC components, waterproof roof and façade, slim construction with distinctive design. |
| 2014 | Cladding UHPCpanels | Foundation Louis Vuitton, France | Creative design |

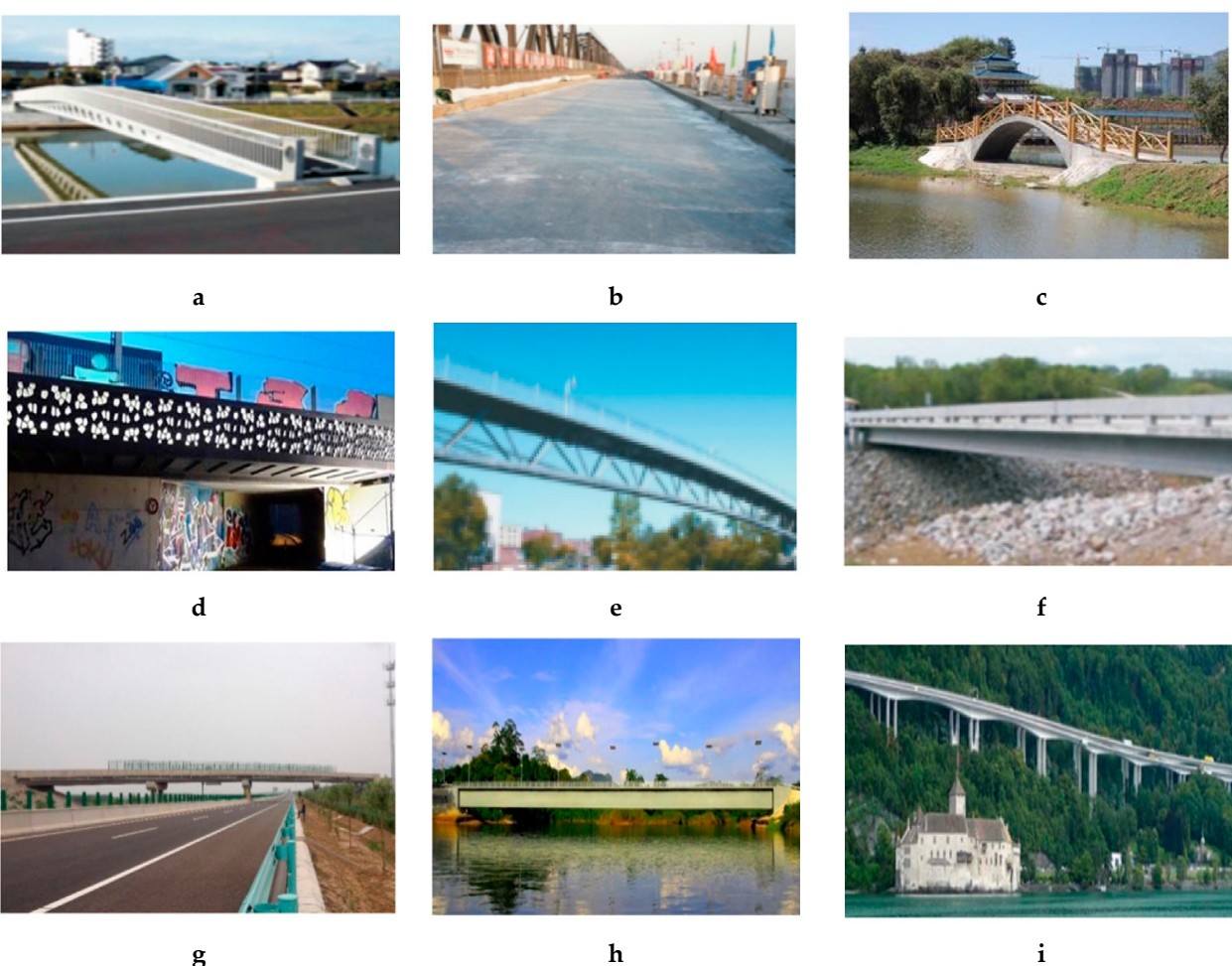

**Figure 3.** Typical apps of UHPC in BE: (**a**) Sakata-Mirar foot BE; (**b**) Zhaoqing Mafang BE; (**c**) ultra-high-performance, fiber-reinforced concrete arch BE in Fuzhou University campus viaducts; (**d**) Martinet foot BE; (**e**) Sherbrooke pedestrian BE; (**f**) Mars Hill BE; (**g**) Shijiazhuang to Cixian highway BE; (**h**) Batu 6 BE; (**i**) Chillon [53,91–94].

UHPCs can be utilized in BE decks, girders, arch rings, and other elements. Table 3 provides a brief overview of the utilization of UHPCs in BE components. In the sections below, the authors highlighted the most common benefits of UHPC in BEs that have been investigated in the literature.

**Table 3.** Applications of UHPC in BE elements.

| Ref. | Year | Nation | Name | App Location | Structure Type | Achievement of Utilizing UHPCs |
|------|------|--------|------|--------------|----------------|-------------------------------|
| [91] | 2011 |  | Zhaoqing Mafang BE | BE deck | Simply supported steel composite beam BE | The first time a UHPC deck was paired with a steel box girder to create a lighter composite girder BE. |
| [93] | 2015 | China | Ultra-high-performance, fiber-reinforced concrete arch BE | Arch ring | Arch BE | To fulfill the strength requirements of the arch ring, which would be exposed to an anticipated CS of more than 100 MPa. |
| [94] | 2015 |  | Shijiazhuang to Cixian highway BE | Girder | Three continuous box girders with multi-span structure | To raise the ultimate strength of the box girder while decreasing its self-weight. |

**Table 3.** *Cont.*

| Ref. | Year | Nation | Name | App Location | Structure Type | Achievement of Utilizing UHPCs |
|------|------|--------|------|--------------|----------------|-------------------------------|
| [95] | 2016 | Malaysia | Batu 6 BE | Whole superstructure | Single-span box girder BE | To address the need for international transportation. |
| [91] | 1997 | Canada | Sherbrooke pedestrian BE | BE deck | Space truss girder BE | To minimize the BE's self-weight and improve its corrosion resistance. |
| [92] | 2014/2015 | Switzerland | Chillon viaducts | Deck slab | Dual-box girder structure | To advance the BE's durability and girder stiffness and the fatigue performance of the slabs. |
| [92] | – | | Martinet foot BE | Girder | A U-formed girder with a simply supported structure | To avoid damage from hazardous fluids and to maintain a crack-free condition under service stress. |
| [91] | 2006 | USA | Mars Hill BE | I-girder | Pre-stressed beam BE | For improved lifecycle and durability |
| [91] | 2002 | Japan | Sakata-Mirar foot BE | Box girder | Pre-stressed simply supported beam BE | To provide design guidance for the UHPC structure in Japan. |

### 2.1. Bridge Piers/Column

The BE pier is a crucial component in BEs since it transfers dead and live loads from the superstructure to the foundations. The box section, a typical pier form in BEs, may efficiently reduce the inertia force in the pier by minimizing its self-weight. Nevertheless, the piers with box-formed cross-sections were damaged in numerous earthquakes, including the 1999 Chichi earthquake [96] and the 2008 Wenchuan earthquake [97]. This is primarily because of the limited lifecycle and ductility of normal-strength concrete piers. Ren et al. [98] evaluated the effectiveness of UHPC box piers exposed to seismic load utilizing computational and empirical methods. The authors investigated the ductility of UHPC box piers with varying longitudinal reinforcement and axial load ratios.

Furthermore, the UHPC was utilized as a pier jacket to improve the concrete BE pier's seismic performance, corrosion resistance, and spalling resistance. As indicated in Figure 4, one of the UHPC pier jacket projects (the Mission BE) was completed in Canada in 2014 [99]. Additionally, including pre-fabricated segmental BE columns and ultra-high-performance, fiber-reinforced concrete (UHPFRC), segmented components of BE structure displayed substantially greater energy dissipation capacity dynamic characteristics and impact resistance [100].

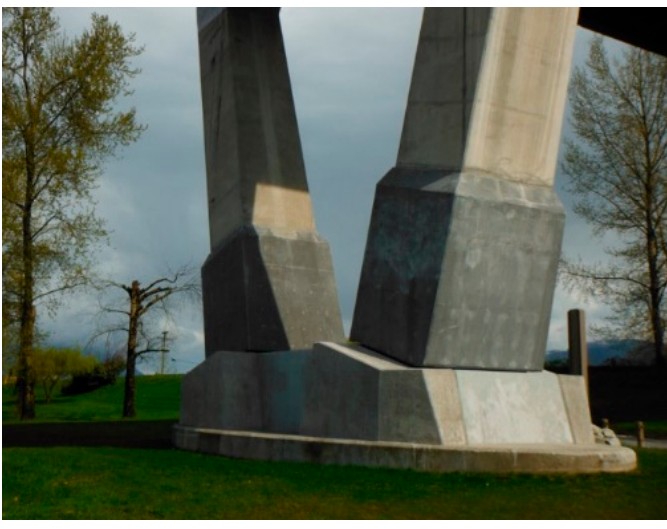

**Figure 4.** Representative apps of UHPC in BEs: pier jacket [99].

In China, a commonly utilized structural method for the construction of BE piers is the concrete-filled steel tubular column, which may also be utilized for the production of multistory buildings and other supporting constructions with appropriate flexibility and high axial strength [101]. With the incorporation of UHPC into BE construction, substantial investigation into the efficiency (ductility and axial capacity) of ultra-high-performance concrete-filled steel tube columns has been conducted [102]. Ultra-high-performance concrete has also been utilized to effectively repair severely damaged reinforced concrete (RC) columns in BE piers [103]. According to the empirical findings, utilizing an ultra-high-performance concrete jacket to repair RC columns substantially advances the cyclic load resistance, energy dissipation capacity, ductility, stiffness, and strength of the column test sample.

Despite its benefits in ductility and strength, the relatively expensive costs of utilizing UHPC in BE piers have restricted its widespread adoption. As a result, attempts have been made to develop optimal UHPC mixture designs for BE piers. According to the investigation by Joe and Moustafa [104], utilizing UHPC may minimize the cross-sections of BE piers, reducing total cement and material use. According to Aboukifa et al. [105], constrained ultra-high-performance concrete cylinders with substantial longitudinal reinforcement ratios would be required to minimize the size of the cross sections and fully exploit the greater strength of UHPC. Although their findings confirmed the viability of utilizing UHPC instead of standard concrete in the construction of significant BE substructures, the cost savings that may be gained by utilizing UHPC for BE piers remains unknown owing to a lack of comparison data.

The hydration heat of UHPC can be significantly high compared to normal concrete due to its high Portland cement dose. Hence, the use of pozzolanic materials in UHPC production, such as fly ash, ground granulated blast furnace slag, etc., reduces the Portland cement dosage, minimizing hydration heat while also addressing engineering and ecological issues [33].

### 2.2. Bridge Piles

The utilization of piles to sustain BE loads is a standard approach for producing high-performance infrastructure. Typically, piles are built utilizing pre-cast concrete, steel sections, or cast-in-place concrete. Concrete piles face challenges such as pile collapse during installation, limited capability, and pile degradation caused by ecological threats.

The utilization of UHPC mixtures in pile fabrication has substantially enhanced the long-term capability performance of the pile. UHPC mixtures with high strengths may be driven with little to no damage. As UHPC mixtures have relatively low permeability, they are more resistant to ecological challenges. The high material costs of UHPC mixtures are balanced by the decreased material required owing to the lower demand and smaller pile sizes for maintenance work over the construction project's lifecycle.

In accelerated bridge construction (ABC) apps, UHPC mixtures are formed into various BE elements for accelerated BE construction, utilizing the prefabricated BE elements system (PBES) tech and prefabricated BE elements. Owing to the application of UHPC in the ABC approach, BE design engineers were able to create new BE sections with geometrical dimensions, leading to the simplicity of construction and considerable material savings. The reformed Pi-girder, advanced by scientists at the Massachusetts Institute of Technology (MIT) and tested at the FHWA's Turner Fairbank Highway Research Center (TFHRC) in McLean, Virginia, is an instance of the creative UHPC prefabricated sections utilized in ABC processes [106].

Moreover, according to a review of the published literature, UHPC's high durability and material characteristics make it a good material for deep foundation applications. UHPC piles have a larger load capacity than steel piles, as shown by the load testing, which should result in a decrease in the number of piles needed for a standard bridge foundation [107].

*2.3. Bridge Decks*

Numerous deterioration-related issues such as delamination cause the end of the lifecycle in BE decks, spalling, and cracking [108], which may also be impacted by temperature and load impacts [109]. UHPC is developing as a novel strategy for a wide range of construction, BE design, and restoration issues. One novel utilization of UHPC in BEs is the rehabilitation of BE decks with the utilization of thin, bonded UHPC overlays.

UHPC is now employed to pour BE deck overlays to enhance BE deck conditions. There is a significant need for the effective and long-term rehabilitation of BE decks that have deteriorated owing to improved vehicle weight and traffic, deck cracking, freeze–thaw cycles, reinforcing steel corrosion, and concrete cover delamination. The available budget and the expected lifecycle of the repaired structure determine the BE deck restoration tech. Historically, specific asphalt mixtures, ordinary concrete overlays, and latex-modified concrete comprising polymer compounds were utilized. UHPC overlays are now being utilized effectively. UHPC overlays are helpful since they allow for the utilization of thin (2.5–5.0 cm) overlays with excellent binding to existing concrete. UHPC's extremely incredible strength and low permeability characteristics offer appropriate strengthening as well as protection against contaminator ingresses such as deicing salts and chemical assaults.

With little extra deadweight on the BE structure, ultra-high-performance concrete overlays can reinforce the structure and avoid future water penetration and chloride [110]. One of the successful uses of ultra-high-performance concrete overlays in BE decks has been the enhancement of the Chillon viaducts, two parallel concrete highway BEs erected in Switzerland in the late 1960s. To protect the safety of the Chillon viaduct, which had been damaged owing to alkali aggregate reactivity, a 45 mm layer of Ultra-high-performance, fiber-reinforced concrete was installed on top of the deck slab in 2015, acting as the deck slab and the main girder's external tensile reinforcement. As a result, the deck slab's stiffness and the ultimate load capability were enhanced, and water penetration into the existing concrete was successfully avoided [111].

Owing to its large capability, low self-weight, and ease of construction, an orthotropic steel deck (OSD) is a popular BE deck tech for long-span BEs. However, numerous investigations have been conducted to address the problem of orthotropic steel decks, which is that they may be subject to fatigue cracking, particularly at the weld between the longitudinal stiffeners and the deck plate [112]. This is primarily owing to the fact that when exposed to traffic pressures, a substantial number of stress cycles are created in the fatigue-prone details [113]. The current investigation focuses on deploying a UHPC layer to rehabilitate existing orthotropic decks to solve the fatigue difficulties associated with orthotropic steel decks. For instance, Shao, Qu, Cao, and Yao [113] presented a lightweight composite deck (LWCD) built from an orthotropic steel deck with a UHPC overlay. According to the scientists, the lightweight composite deck had improved local stiffness and might efficiently prolong the fatigue life of the BE deck. Yoo and Choo [114] built a deck with an ultra-high-performance, fiber-reinforced concrete layer connected to steel girders.

Furthermore, the deck pavement is a direct-wear part of the BE structure that is impacted not only by vehicle wear but also by thermal expansion and rain erosion. As the lifespan of the BE deck pavement in China has decreased considerably in recent years because of the continual rise in traffic loads, some scientists are considering adopting UHPC materials instead of standard paving materials to counteract this challenge. Table 4 illustrates instances of the UHPC BE deck pavement utilized in China in recent years.

**Table 4.** Applications of UHPCs in BE deck pavement in China.

| Position | Year | Name | BE Type |
|---|---|---|---|
| Guangdong Province | 2011 | Ma Fang BE | Simple box girder |
| Guangdong Province | 2014 | Buddha Chen BE | Variable section continuous steel box girder |
| Hunan Province | 2015 | Dong Ting Lake Second BE | Plate-truss composite suspension BE |
| Beijing | 2015 | Tong Hui BE | Deck beam arch combination BE |
| Tianjin | 2015 | Hai, He BE | Hybrid beam cable-stayed BE |
| Guangdong Province | 2016 | Rong Jiang BE | Hybrid beam cable-stayed BE |

*2.4. Bridge Girders*

BEs are subjected to higher live loads than buildings; thus, as shown in Figure 5, UHPC has proven to be a viable material for BEs. The first UHPC BE—the Mars Hill BE, as shown in Figure 5a—in the United States was advanced in 2006 in Wapello County, Iowa. UHPC was employed to build the 33.5 m long, I-formed pre-stressed BE girders. UHPC's greater CS aided in reducing the girder depth and numerous pre-stressing tendons [115]. Depending on the project's success, the form of the UHPC girder was further improved, and Pi-formed girders were constructed, as seen in Figure 5b. After it was opened to traffic, the Jakway Park BE performed well because of its optimum design [116]. The improved productivity of a Pi-formed UHPC girder has significantly broadened the utilization of UHPC in BEs. However, the high initial costs of the proprietary UHPC employed in the two projects precluded UHPC from being widely adopted in other states. As a result, non-proprietary, cost-effective UHPC has been developed utilizing locally accessible materials. However, the potential for the employment of non-proprietary UHPCs as structural BE parts has to be researched further.

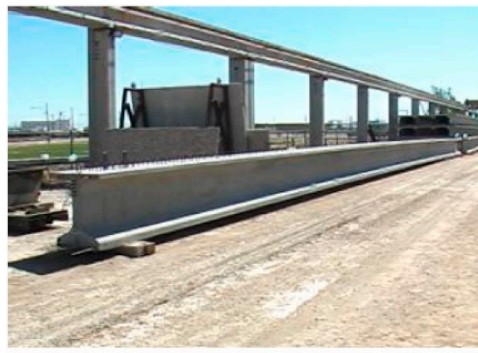 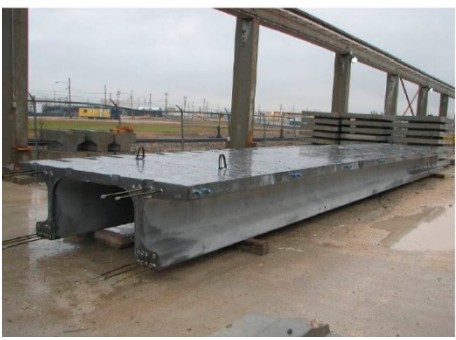

**a**            **b**

**Figure 5.** Representative apps of UHPCs in BEs: (**a**) I-formed girder [115]; (**b**) Pi-formed girder [116].

The component strength at the strength limit state is primarily considered in the design of an ultra-high-performance concrete girder [117]. The corrosion of the girder ends can dramatically decrease the strength of the BE components [118], highlighting the necessity for material with strong resistance to the infiltration of hazardous chemicals. Previous investigation has confirmed UHPC's improved corrosion resistance. However, owing to the high costs of UHPC, its utilization in BE girders has mostly been as a repair material. UHPC, for instance, has been utilized to cover corrosion-damaged steel girder ends [118]. In this unique repair procedure, ultra-high-performance concrete panels were connected to girders utilizing shear studs welded to the flange and web around the corroded region. According to the findings of this research, employing UHPC can recover the capability lost because of corrosion damage. Graybeal performed a series of investigations on the flexural behavior of ultra-high-performance concrete BE girders and determined that UHPC

I-girders might have a larger flexural capability than normal strength concrete girders with the same cross-sectional geometry [119].

UHPC girders have the added benefit of being lighter in self-weight, which might be employed in steel–concrete composite BEs to minimize production costs in long-span and continuous BEs. For the trial design of the steel ultra-high-performance concrete lightweight composite BE at the South Dongting Lake, BE, Shao and Hu [120] presented a set of novel steel–UHPC composite girders. The self-weight of the steel–ultra-high-performance concrete composite girder could be lowered by about 35 percent when compared to the conventional composite girder.

### 2.5. Long-Span Bridge

The Seonyu foot BE in South Korea, with a main span of 120 m, was fabricated utilizing ultra-high-performance concrete in 2002 and finished in 2004 [121]. The Seonyu foot BE structure, the world's longest long-span BE, built utilizing ultra-high-performance concrete, required around half the material volume that would have been utilized in conventional concrete production while providing similar strength characteristics. In Japan, the Sakata-Mirai foot BE, with a span of fifty meters, was built in 2003. The BE demonstrated how a perforated web in an ultra-high-performance concrete superstructure can decrease weight while still being visually beautiful [122]. Subsequent to the achievement of these projects, ultra-high-performance concrete pedestrian BEs have been constructed throughout Australia, Europe, Asia, and North America (Canada and the U.S.) [123].

At the moment, some forms of flaws in long-span BEs have appeared, including (1) steel BE deck pavement and BE surface structure crack; (2) general crack and deflection of pre-stressed concrete continuous box girder; and (3) concrete crack in the negative moment area of steel–concrete composite beams [63,71].

The impact of the long-span BE's bent and broken concrete beam has been a severe concern. Scholars have suggested the one-way, pre-stressed UHPC thin-wall continuous box girder construction to tackle these challenges. The conceptual design for the UHPC continuous box girder BE been completed. According to the research, this innovative form of UHPC construction could effectively minimize girder BE deflection and fracture. Using UHPC with a high tensile strength as the BE system instead of traditional concrete could dramatically enhance the stiffness of the BE deck, substantially advance the stress of the pavement layer, effectively minimize cracking, and decrease the fatigue stress of the steel structure [69].

In recent decades, the construction and design tech for long-span BEs has evolved at an increasing rate. Innovations in structural construction and design are often associated with utilizing the novel product, and UHPC has a lot of promise in this area.

### 2.6. Joints/Links

All aspects of the BE, including the joints, must be considered (for example, expansion joints, transverse and vertical wet joints, etc.) to achieve a long-lasting and safe highway traffic system. Table 5 describes the UHPC apps in BE joints.

The research shows that link slab components could be included in BE decks to replace the expansion joints.

This is often utilized in ABC projects. The standard connection requires complex reinforcing arrangements, which take time. The utilization of UHPC simplifies the on-site reinforcement and reduces construction time and assembly procedures. As indicated in Figure 6, two field-cast UHPC connection projects (Route 23 BE in Oneonta and Route 31 BE in Lyons) were built in 2009.

**Table 5.** Applications of ultra-high-performance concrete in BE joints.

| Ref. | Year | Nation | Name | App |
|------|------|--------|------|-----|
| [124] | 2011 | | Fingerboard Road BE | Joints among deck bulb tees |
| [125] | 2011 | U.S. | U.S. Route 6 BE | Transverse and longitudinal joints among beams |
| [126] | 2016 | | Pulaski Skyway | Joint fill connections among the sheer pockets and slabs |
| [125] | 2007 | | Sunshine Creek BE | Joint fill among pre-cast curbs and adjacent box beams |
| [127] | 2009 | | Buller Creek BE | Joint fill among adjacent box beams and among pre-cast curbs |
| [127] | 2009 | | Eagle River BE | Joint fill among pre-cast curbs and among adjacent box beams to establish live load continuity |
| [127] | 2010 | Canada | Wabigoon River BE | Joint fill among pre-cast curbs and among adjacent box beams |
| [126] | 2013 | | Blackwater River BE | Joint fill among pre-cast curbs and among adjacent box beams |
| [126] | 2016 | | Nipigon River BE | Connections of pre-cast tower segments to connections of longitudinal, cast-in-place tower segments and transverse joints to steel beams and girders |

Chloride-contaminated water infiltration and debris deposition issues connected to expansion joints often occur, posing a significant long-term repair and maintenance work challenge and impacting on the structural lifecycle of BEs [128]. BE expansion joints are expensive to build and maintain. To overcome this issue, jointless and continuous BE decks with link slabs connecting adjacent girders have been suggested and advanced [129].

Several investigations [130–132] found that ductile concrete materials, known as engineered cementitious composites (ECC), with ordinary steel reinforcement might be employed to produce engineered cementitious composite link slabs. This connection slab was designed to resist the imposed bending moment caused by the relative deck rotations [133] or traffic loads [134]. Further investigation [135] found that ECC-linked slabs strengthened with non-corrosive fiber-reinforced polymer rather than steel reinforcement may greatly improve crack width control, deflection capability, and corrosion resistance.

Owing to their high ductility and strength, UHPCs could be a desirable material in link slab apps [136], as seen in Figure 6. The usage of an ultra-high-performance concrete connection slab is detailed in Graybeal's work [137], as illustrated in Figure 7. The UHPC link slab in Figure 8 was utilized to repair the leaky joint, providing deck continuity and a long-lasting seal. This link slab has been functional since 2013. The performance of ultra-high-performance concrete jointless BE decks is unknown; therefore, the design could include ambiguities. Regardless of the various advantages of jointless BE decks, the United States has a standardized set of design standards and processes for such BEs, with just a few specifications and design guidelines provided.

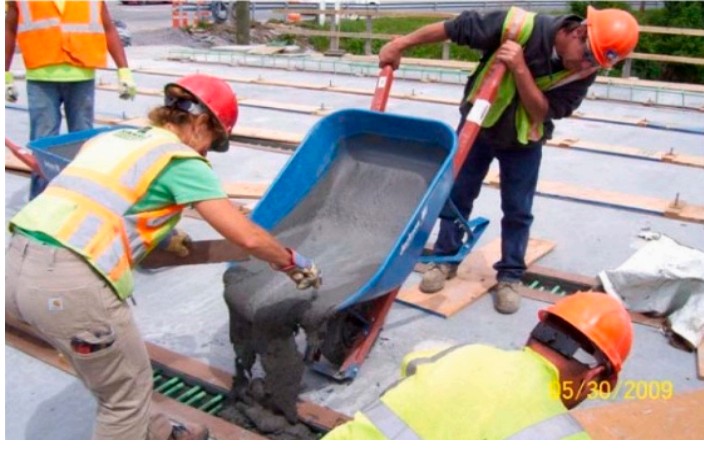

**Figure 6.** UHPC link slab utilized among girders on BE decks [136].

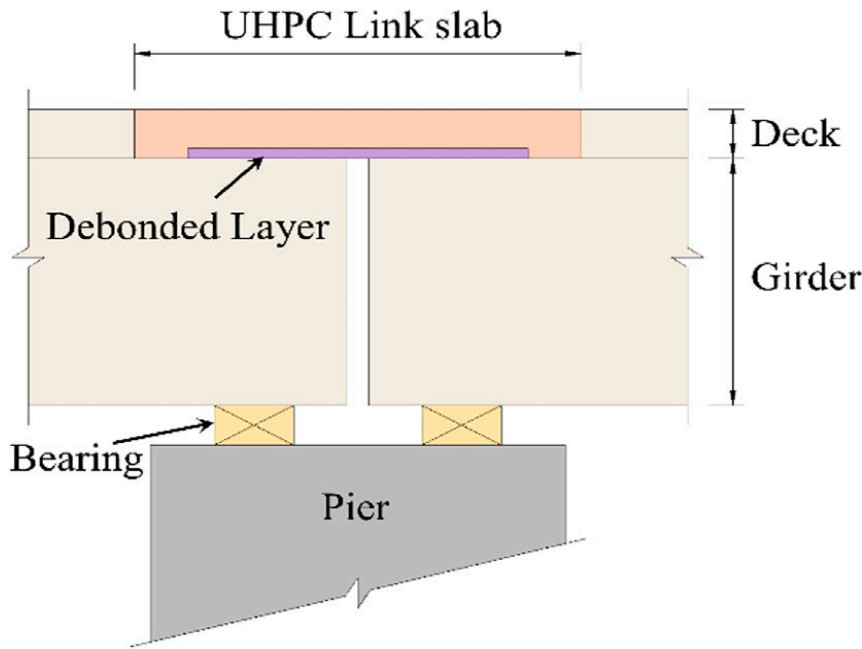

**Figure 7.** Elevation image of the UHPC link slab [137].

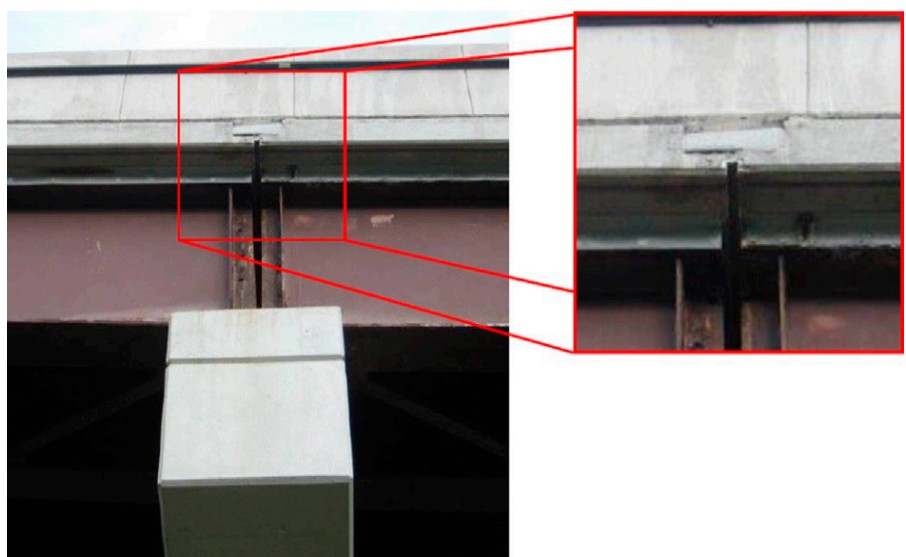

**Figure 8.** Representative apps of UHPC in joint connection BEs [138].

### 2.7. Bridge Bearing Component

Regarding bridge bearings, not only must high bearing compression forces be transmitted but also rotation and movement [139]. A significant manufacturing effort is required to fabricate the form in steel for spherical bearings. Using UHPC instead of steel might well have considerable advantages. Such bearing advantages are obtained without compromising the bearing's service performance or load resistance capacity. Moreover, Table 6 summarizes and briefly analyzes the usual BEs in the app example.

**Table 6.** Examples of UHPC apps in BE bearing components.

| Ref. | Year | Country | Name | App Location | Purpose of Utilizing UHPC |
|---|---|---|---|---|---|
| [140] | 2015 | China | Fuzhou University Landscape BE | Arch rib | Empirical BE to promote the utilization of UHPC |
| [141] | 2007 | Canada | Glenmore Pedestrian BE | Pre-stressed T-beam | Weather resistance and ease of maintenance-work. |
| [142] | 1997 | | Sherbrooke Overpass | Pre-stressed, post-tensioned space truss | To investigate novel materials and architectures. To improve the longevity of ecological compatibility. |
| [143] | 2008 | U.S. | Jakway Park BE | Pi-formed beam | To provide direction for future designs that utilize UHPC Pi-girders. |
| [144] | 2008 | | Cat Point Creek BE | I-formed beam | To use material tensile characteristics to make building simpler. |
| [145] | 2006 | | Mars Hill BE | I-formed beam | To investigate UHPC characteristics and enhance their materials. |
| [146] | 2008 | France | Pont du Diable Pedestrian BE | U-formed beam | To improve the span length and strive for a light, beautiful design. |
| [147] | 2007 | | Pinel BE | Pre-stressed T-beam | To use UHPC for durability and rapid building. |
| [148] | 2005 | | PS34 BE | Box girder | To change the BE's design and incorporation with the surrounding environment and lighten the structure. |
| [148] | 2005 | Australia | Shepherds Gully Creek BE | Pre-cast, pre-tensioned I-beam | Empirical BEs to replace the old, aging timber BE and enhance the bearing capability. |
| [149] | 2007 | Germany | Friedberg BE | Pi-formed beam | To use superior durability characteristics to replace an old, deteriorated timber structure. |
| [150] | 2010 | Austria | WILD BE | Arch rib | Ecological management and light and slim constructions. |
| [148] | 2002 | South Korea | Peace BE | Pi-formed beam | To strengthen diplomatic connections with France while also improving arch performance. |
| [151] | 2009 | | Office Pedestrian BE | Cable-stayed BE | Lightweight structure and reasonable stress |
| [152] | 2013 | Czech Republic | Celakovice Pedestrian BE | Segmental deck | Reduced lifecycle costs and low maintenance-work. |

## 3. Limitations

UHPC, as a cementitious engineering material, provides durability, higher strength, and a compact microstructure, leading to its expanded utilization in infrastructure construction [153]. Most of the time, UHPC fails to fulfill engineering performance standards and is excessively expensive when compared to conventional concrete mixtures [5]. Compared to traditional-strength concrete, UHPC provides superior durability and mechanical performance; nevertheless, its application is limited because of its high initial costs, high cement content (up to 1100 kg/m$^3$), limited design standards, and substantial ecological impact. Furthermore, the high costs of raw materials such as steel fibers and fine silica sand, which accounted for nearly half of the overall costs, restricted its application [154].

The primary limits for utilizing UHPC in BE:

i.  Costs of raw materials: The most significant aspect for BE designers and owners is that numerous raw ingredients (such as steel fiber and silicon) and the costs of raw UHPC materials are more costly than those of standard concrete.

ii.  Ecological sustainability: The production of one tonne of PC releases about the same volume of carbon dioxide into the atmosphere as the burning of one tonne of coal [155], putting a significant burden on ecological sustainability.

iii.  Raw material requirements: UHPC is highly strict with regard to the utilization of raw materials; the form of fiber, the diameter of the gravel, and the water reducer influence the completed product's performance. As a result, how to prepare UHPC for stable performance under varying situations has become the critical challenge limiting its widespread deployment.

iv.  Ultra-high-performance concrete opposes the current objective of sustainable production, which is to reduce greenhouse gas emissions and energy consumption [156]. Furthermore, depending on the usual empirical findings, partial or total strength loss in ordinary or high-strength concrete is more likely to develop as substitution rates rise [156]. Achieving a higher replacement level for concrete mixes without losing the hardened characteristics of concrete remains a fundamental problem in developing optimum UHPCs in terms of mechanical functioning and sustainability.

v.  Specifications: In the meantime, appropriate, standardized rules and standards for design, testing, numerical modeling, and construction should be developed like those of ordinary concrete. Furthermore, before the large-scale deployment of UHPC materials, procedures for adequate maintenance work, recognizing damage, and replacing or repairing UHPC components must be advanced and standardized to facilitate UHPC apps.

vi.  Maintenance-work requirements: In order to achieve high material strength, UHPC requires high-temperature maintenance work throughout construction. However, the BE construction process may not always be equipped with the necessary equipment for such maintenance work. As a result, UHPC is usually utilized in the prefabricated form, which limits the options for BE designs and building techniques.

In conclusion, practical apps for ultra-high-performance concrete can be found worldwide. UHPCs, oppositely, are going slowly, with constraints restricting their apps. High initial costs, restricted codes, design difficulties, sophisticated manufacturing techniques, and limited available resources impeded its commercial growth and deployment in modern construction, particularly in emerging nations. To fully realize UHPC's enormous potential, the sector must improve its collaboration with governmental organizations, academic institutions, and owners. All parties should share this novel practical experience and knowledge of the material. As UHPC is a material with high material sensitivity, local design standards and guidelines should be advanced. More investigation into developing cost-effective and sustainable UHPC, utilizing alternative materials with comparable functionality to replace the costly UHPC composites while minimizing the ecological impact is required for improved UHPC adoption. Engineers, architects, and designers should be more willing to experiment with novel technologies and materials. With all of these efforts, UHPC could become a current and future building material, providing a more comprehensive strategy for sustainable construction.

## 4. Challenges

In various nations through the past two decades, ultra-high-performance concrete has been utilized for both structural and non-structural pre-cast elements. Nevertheless, owing to a lack of design guidelines and its high initial costs, this significant tech has failed to become a popular tech for daily use. Moreover, the high energy consumption and high costs of UHPC materials make it problematic for it to compete with CC designs, limiting its usage. Investigation into enhancing the sustainability and the lowering of the costs of ultra-high-performance concrete is needed to enable its future widespread use. Numerous investigations were conducted to modify the material mixtures utilizing

industrial by-products and local raw materials to decrease the proportion of PC, SP, and steel fiber [157]. With reduced ecological and cost influences, ultra-high-performance concrete will be much simpler for the infrastructure market to adopt and will arouse the attention of infrastructure owners.

The primary challenges of utilizing UHPC in BE are as follows:

1.  Few investigations have been conducted on the lifecycle assessment of UHPC structures. The construction industry benefits from UHPC owing to its excellent mechanical and durability characteristics. UHPC constructions have a longer lifecycle, need less maintenance work, and have lower repair costs than typical concrete structures [158]. This must be considered throughout the design phase of the structure.

2.  As UHPC structures differ from normal RC requirements and the number of engineers, skilled builders, and experts are restricted, the teams experienced with UHPC design and tech challenges are necessary In the UHPC market, there are only around five major companies, mostly in North America and Europe [110]. Standards for the construction and design of UHPC buildings must be devised depending on empirical research, prior knowledge, field experiences, and scientific computation. International guidelines are difficult to develop owing to the wide range of UHPC experience in various nations [85].

3.  UHPC materials are energy-intensive and costly, restricting their use. UHPC requires more study to reduce the costs and improve long-term sustainability. Several investigation studies modify material mixes by utilizing industrial by-products and regional raw materials in order to minimize cement, SP, and steel fiber consumption. If the ecological and economic costs and the effects of UHPC are decreased, infrastructure owners will be more interested.

4.  Lack of knowledge about mixing, quality control, and synthesis procedures is a challenge since UHPC mixes with steel fibers and needs a multi-step mixing process and a special curing method [99].

5.  Owing to the rapid curing and high binder dose, creep and shrinkage have a substantial impact on UHPC behavior. More study is required to investigate materials at the nano-, micro-, and macro-levels to correlate structural behavior and physical phenomena for large-scale building methods.

## 5. Future Prospects

With regard to the above review, the following future investigation topics have been highlighted:

1.  The static and dynamic behavior of BE connections and elements/components made of UHPC materials is fundamentally modeled. The models can be utilized in widely accessible commercial software (For instance, ANSYS, SAP2000, etc.).

2.  Develop a design and construction approach for pre-stressed UHPC girders for long-span BEs.

3.  Optimal performance and reliability design methodologies include the complete lifecycle costs of a BE, including design, construction, maintenance work, and retrofitting for damaged components caused by severe occurrences such as earthquakes, hurricanes, vessel collisions, etc.

4.  The enhanced lightweight UHPC can be employed to make portable BE deck panels [31].

5.  Despite investigations demonstrating the possibility of substituting normal-strength concrete with UHPC in BE apps, the strategies for lowering the UHPC costs have not been extensively established, emphasizing the need for more investigations on this topic to broaden and extend UHPC uses in BEs.

6.  To broaden the applicability of UHPC to jointless BEs, further investigations into the failure mechanism and mechanical characteristics of UHPC utilized for link slabs are required.

## 6. Summary

UHPC is an innovative novel material with superior characteristics, such as remarkable strengths and good durability, obtained via improvements in packing and homogeneity density. Since its debut in the early 1990s, there has been a substantial accumulation of knowledge on the construction, design, and material of UHPC structures, with several nations attempting to apply it to building and BE apps. UHPC materials exhibit high mechanical characteristics and durability, which can enhance the connection integrity of BE component joints, improve BE load-carrying capability, and minimize BE pavement cracking and deformation. They are presented as exclusively utilized in small- and medium-sized BEs, as well as pedestrian BEs.

UHPC is anticipated to be employed to address a variety of issues, including the general cracking and deflection of conventional pre-stressed concrete continuous steel structures, damage to steel deck pavement, fatigue cracks in steel structures, concrete cracks in the negative moment area of steel–concrete composite beams, and a variety of difficult issues associated with long-span BEs. The utilization of pre-cast segmental BE columns has been actively promoted in engineering procedures. Although conventional segmental RC columns are self-centering, their damage tolerance and energy dissipation capability remains restricted. According to the authors' empirical and numerical analyses, pre-cast segmental UHPC columns have recently been advanced and can easily solve the aforementioned restrictions. Although the cyclic response of such BE columns has been studied empirically, the impacts of pre-cast segmental UHPC columns on the overall performance of a BE at the system level are still unknown. An extensive study on UHPC preparation procedures, structural design approaches, material characteristics, and requirements will lead to decreased material costs and broader apps.

**Author Contributions:** Conceptualization, S.A., W.M., I.A. and M.N.; methodology, A.A., Y.O.Ö. and M.H.A.; software, S.A.; validation, S.A., W.M., I.A., M.N., A.A., Y.O.Ö. and M.H.A.; formal analysis, A.A.; investigation, S.A., W.M., I.A., M.N., A.A., Y.O.Ö. and M.H.A.; resources, M.H.A.; data curation, M.H.A.; writing—original draft preparation, S.A., W.M., I.A., M.N., A.A., Y.O.Ö. and M.H.A.; writing—review and editing, S.A., W.M., I.A., M.N., A.A., Y.O.Ö. and M.H.A.; visualization, S.A.; supervision, M.H.A.; project administration, S.A.; funding acquisition, M.H.A. All authors have read and agreed to the published version of the manuscript.

**Funding:** This research received no external funding.

**Data Availability Statement:** Not Applicable.

**Conflicts of Interest:** The authors declare no conflict of interest.

## Abbreviations

| | |
|---|---|
| BE | Bridge engineering |
| UHPC | Ultra-high-performance concrete |
| UHPFRC | Ultra-high-performance, fiber-reinforced concrete |
| CS | Compressive strength |
| RC | Reinforced concrete |
| ABC | Accelerated Bridge Construction |
| SF | Silica fume |
| OSD | Orthotropic steel deck |
| FHWA | Federal Highway Administration |
| LWCD | Lightweight composite deck |

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
