# Peer review of "Application of Ultra-High-Performance Concrete in Bridge Engineering: Current Status, Limitations, Challenges, and Future Prospects"

_buildings, doi:10.3390/buildings13010185_

Round 1

Reviewer 1 Report

1. Abbreviations of SIFCON and CRC should be expanded first.

2. The expansion of RPC should be done first

3. Page 5 - Line 25 - "sustainable way of producing UHPC may be done with GPC technique" may be included with the following reference:

https://doi.org/10.1016/j.jobe.2021.102267

4. Data in Table 1, 3, 4, 7 can be provided with year-wise order.

5. As the size of the bridge piers/columns are huge, what about the heat of hydration effects in UHPC with huge quantum of cement. Discussion is required for this case.

6. Title of section 2.5 may be provided with "Long-span Bridge".

7. The numbering in Section 4 and 5 may starts with 1.

8. Overall, grammatical checks are required.

Author Response

Dear Reviewer,

Thanks for all the time and effort you put into reviewing our manuscript. This is a great contribution to the scientific community. It's much appreciated. Thanks.

Regarding the comments, you will find the authors' responses in the attachments. Also highlighted in the revised manuscript.

Kind Regards,

Authors

Reviewer 2 Report

This review paper deals with the Application of Ultra-High-Performance Concrete in Bridge Engineering. The authors have performed a detailed review of the application of UHPC on existing bridge structures around the world. The manuscript is well-written, and it explains the advantages of the UHPC in bridge engineering.

The reviewer suggests a few points for the betterment of the manuscript.

1. Line 27: Remove “1. Introducion”

2. Check the section numbering, Section 1.1 is missing.

3. Section 1.2 can be included with a few points about the design principles of UHPC.

4. Section 2.7 – A short introduction about the section may be included in order to understand the necessity of the bearing/pedestal component and the advantages of UHPC as such a component.

5. Few components like pier cap, Isolated open footing (foundations) components can also be included to have a complete study on the entire structural components of the bridge structure.

6. The numbering in sections 4 & 5 can start with 1 again.

7. Some details are missing in some references listed in the manuscript (kindly mention if it is a report or thesis). Use a uniform format.

Author Response

(The authors gave the same response as above.)

Reviewer 3 Report

In this paper, the authors reviewed the application of UHPC in bridge engineering. Many research in UHPC has been conducted in recent years, however, very few comprehensive reviews can be found. The authors provides lots of information regarding the past and current research in this area, and also suggested a path forward to better understand this topic and commercialization.

Generally speaking, the paper is well written. The authors should double-check the formatting such as word spacing, numbering, etc. 

Author Response

(The authors gave the same response as above.)
